# Genetically encoded Nδ-vinyl histidine for the evolution of enzyme catalytic center

Haoran Huang[1,2], Tao Yan[1,2], Chang Liu[1,2], Yuxiang Lu[1], Zhigang Wu[1], Xingchu Wang[1] & Jie Wang ®[1] ✉

Genetic code expansion has emerged as a powerful tool for precisely introducing unnatural chemical structures into proteins to improve their catalytic functions. Given the high catalytic propensity of histidine in the enzyme pocket, increasing the chemical diversity of catalytic histidine could result in new characteristics of biocatalysts. Herein, we report the genetically encoded Nδ-Vinyl Histidine (δVin-H) and achieve the wild-type-like incorporation efficiency by the evolution of pyrrolysyl tRNA synthetase. As histidine usually acts as the nucleophile or the metal ligand in the catalytic center, we replace these two types of catalytic histidine to δVin-H to improve the performance of the histidine-involved catalytic center. Additionally, we further demonstrate the improvements of the hydrolysis activity of a previously reported organocatalytic esterase (the OE1.3 variant) in the acidic condition and myoglobin (Mb) catalyzed carbene transfer reactions under the aerobic condition. As histidine is one of the most frequently used residues in the enzyme catalytic center, the derivatization of the catalytic histidine by δVin-H holds a great potential to promote the performance of biocatalysts.

The inherent diversity in genetically encoded chemical spaces facilitates the adaptation of enzymes to recognize new substrates and catalyze unprecedented reactions through mutagenesis and directed evolution[1–3]. However, the relatively limited chemical diversity of the 20 natural amino acids (NAAs), which serve as the fundamental building blocks of enzymes, hinders the precise fine-tuning of the chemical structure of enzyme catalytic centers[4].

In the past two decades, genetic code expansion has emerged as a powerful tool for precisely incorporating unnatural chemical structures into proteins, thereby overcoming the limitation of chemical diversity in NAAs[5–7]. Despite the successful genetic encoding of numerous unnatural amino acids (UAAs)[8,9], which have been utilized to improve reaction scope and enzyme performance[4,10–33], there is a continuous need to expand the chemical diversity of enzyme catalytic residues. Given the prevalence of histidine in enzyme catalytic centers, enhancing the chemical diversity of catalytic histidine has become crucial for introducing novel properties to biocatalysts. Increasing the

chemical diversity of catalytic histidine will improve the properties of biocatalysts[34].

For example, the imidazole group of histidine (His) serves as a nucleophilic group in the catalytic center of hydrolases[35–37] and as a metal-chelating group in the catalytic center of metalloenzyme[38,39]. The development of genetically encoded histidine analogs has enabled the fine-tuning of His-containing catalytic centers[40], leading to improvements in enzyme function. The development of Nδ-methyl histidine (δMe-H)[41] is one of the most successful examples to demonstrate that fine-tuning of the catalytic center can significantly improve enzyme performance. Specifically, δMe-H was utilized to improve the conversion yield[12], TON[21], and reaction scope of esterases[10], MBHases[11], and heme-dependent enzymes[17,19,21,33]. To date, only a limited number of histidine analogs with unique chemical properties have been developed[20,41] and applied in the evolution of biocatalysts.

Herein, we reported the use of the genetically encoded Nδ-vinyl histidine (δVin-H), which features more electron-withdrawing

[1]Department of Chemistry, Research Center for Chemical Biology and Omics Analysis, College of Science, Guangdong Provincial Key Laboratory of Catalysis, Southern University of Science and Technology, Shenzhen 518055, China. [2]These authors contributed equally: Haoran Huang, Tao Yan, Chang Liu. ✉e-mail: wangjie@sustech.edu.cn

imidazole group than native histidine or δMe-H. We first identified a highly efficient pyrrolysine aminoacyl-tRNA synthetase (PylRS) encoding δVin-H as a protein of interest (POI) with satisfactory incorporation efficiency. Subsequently, we introduced δVin-H into the active center of the OE1.3 esterase[11] and further demonstrated that the enzyme-containing δVin-H exhibits better enzyme activity and kinetics than native His and δMe-H, which is consistent with the original design of δVin-H. We further introduced δVin-H into four heme-dependent proteins to replace the axially coordinated His, all of which showed high incorporation efficiency. Finally, we verified that δVin-H facilitated functional improvements in the myoglobin variant Mb* (H64V, V68A)[42]. We found that δVin-H not only improved the yield of the Mb-catalyzed cyclopropanation reaction but also expanded the scope of the substrate to the electron-withdrawing styrene, which was previously ineffective for WT Mb*. Additionally, we explored the Mb*-δVin-H-catalyzed alkylation of silane, revealing an enzymatic conversion of the Myoglobin protein. Collectively, these findings highlight the potential of δVin-H in enhancing enzyme functionality.

## Results

### Design and synthesis of δVin-H

We started with the design of histidine analogs for the evolution of enzyme catalytic centers. In many histidine-involved catalytic processes, the epsilon (ε)-position N atom (Nε) of imidazole typically serves as the active site involved in nucleophilic attack or metal coordination (Fig. 1A). Thus, Nδ-substituted histidine has potential to enhancing the catalytic performance of Nε[34]. Notably, Nδ-methyl His has previously demonstrated success in improving the performance of enzymes such as esterases[10], MBHases[11], and heme-dependent enzymes[12,17,19,21,33]. Here, we designed a vinyl group and added the vinyl group to the Nδ of His, which has a distinct effect on the imidazole ring compared to that of δMe-H. As shown in Fig. 1B, C, the vinyl substituent at Nδ position constructed a more acidic Nε-hydrogen bond and a more electron-withdrawing imidazole ring, thereby creating a unique catalytic center crucial for nucleophilic attack or metal coordination. Furthermore, the pKa of natural amino sidechains typically ranges from 3 to 12[43], which are involved in tuning the structure or function of proteins in response to pH changes. However, as depicted in Fig. 1D, none of the natural residues possess a pKa within the physiologically relevant pH range of 4.5-7. This pKa "blank" may limit the functional evolution of proteins. To address this, we successfully tuned

the pKa of the Nε-hydrogen of histidine from 7.07 to 5.71 via vinyl substitution (verified by titration), thereby filling the pKa "blank" as designed (Supplementary Fig. 1).

Next, we aimed to genetically encode this rationally designed vinyl-imidazole into proteins for functional evolution. We first designed the structure of δVin-H and developed the corresponding synthetic route for δVin-H (Figs. 1E, 2A). Initially, we attempted to use nucleophilic substitution or metal-catalyzed C–N bonding reactions[44] to add a vinyl group to histidine. However, we mainly obtained Nε-vinyl histidine, and only a few Nδ-vinyl histidine moieties were detected in the product mixture (Supplementary Fig. 2). To address this, we introduced a triphenylmethyl group as a protecting group to the Nε position, followed by the addition of a bromo-ethyl group to the Nδ position. Subsequently, we removed the triphenylmethyl group and conducted an elimination reaction to obtain Nδ-vinyl histidine. We next scaled up this reaction and demonstrated that δVin-H could be synthesized and prepared at the gram scale. To confirm that all vinyl groups were modified at the Nδ position, we conducted 2D NMR analysis, and the results convincingly matched the structure of δVin-His rather than εVin-His (Fig. 2B, C).

### Genetic encoding of δVin-H

As the next step of our study, we aimed to evolve a synthase for the recognition of δVin-H (Fig. 3A). As δMe-H has been successfully incorporated into proteins by a PylRS mutant (δMeH-RS)[41], we first tested δMeH-RS to determine whether it could recognize δVin-H. Although δMeH-RS exhibited a recognition capability for δVin-H, its incorporation efficiency was relatively low (Fig. 3C). Therefore, we would like to evolve PylRS based on the δMeH-RS mutation. Inspired by the crystal structure of PylRS (PDB: 3qtc), we selected two sites for site-directed saturation mutagenesis (311-NNK and 313-NNK). After the negative selection process followed by positive selection via a dual reporter system (Fig. 3A), we identified two PylRS mutants (hit #1 and hit #2) for the recognition of δVin-H (Fig. 3B, Supplementary Fig. 3).

To further evolve PylRS to obtain a mutant with wild-type-like incorporation efficiency, we fixed the N311D mutation and introduced saturation mutagenesis at positions 270, 271, 274, and 313 to construct a new library. We next subjected the library to positive selection via a dual reporter system (Fig. 3A), and subsequently identified a new hit (#3) for δVin-H recognition. We further compared the incorporation efficiency of hits #1, #2, and #3 and found that the best variant, hit #3,

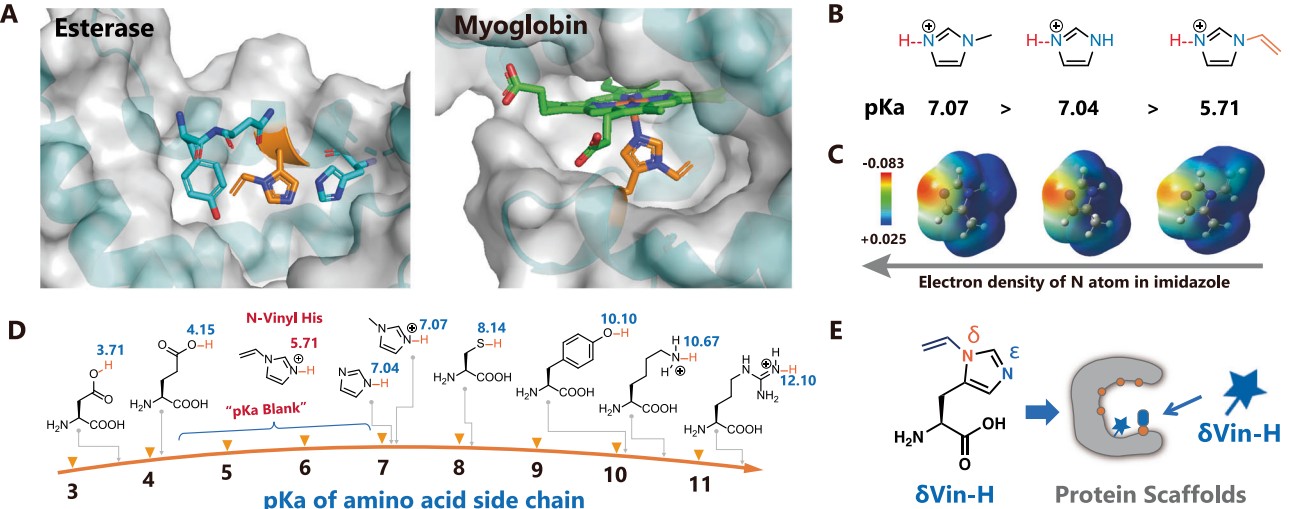

**Fig. 1 | Design of Nδ-vinyl histidine for enzyme evolution. A** A cartoon representation of the His in the catalytic center of esterase- and heme-dependent enzymes was generated using PyMOL (the structures of esterase and myoglobin were generated from the PDB files 6q7o and 5oj9). **B** pKa comparison of imidazole and its derivatives. **C** Comparison of the surface charges of imidazole and its derivatives. **D** pKa ranges of natural amino acid sidechains and imidazole derivatives. **E** Scheme of the evolution of enzymes enabled by Nδ-vinyl histidine. Source data are provided as a Source Data file.

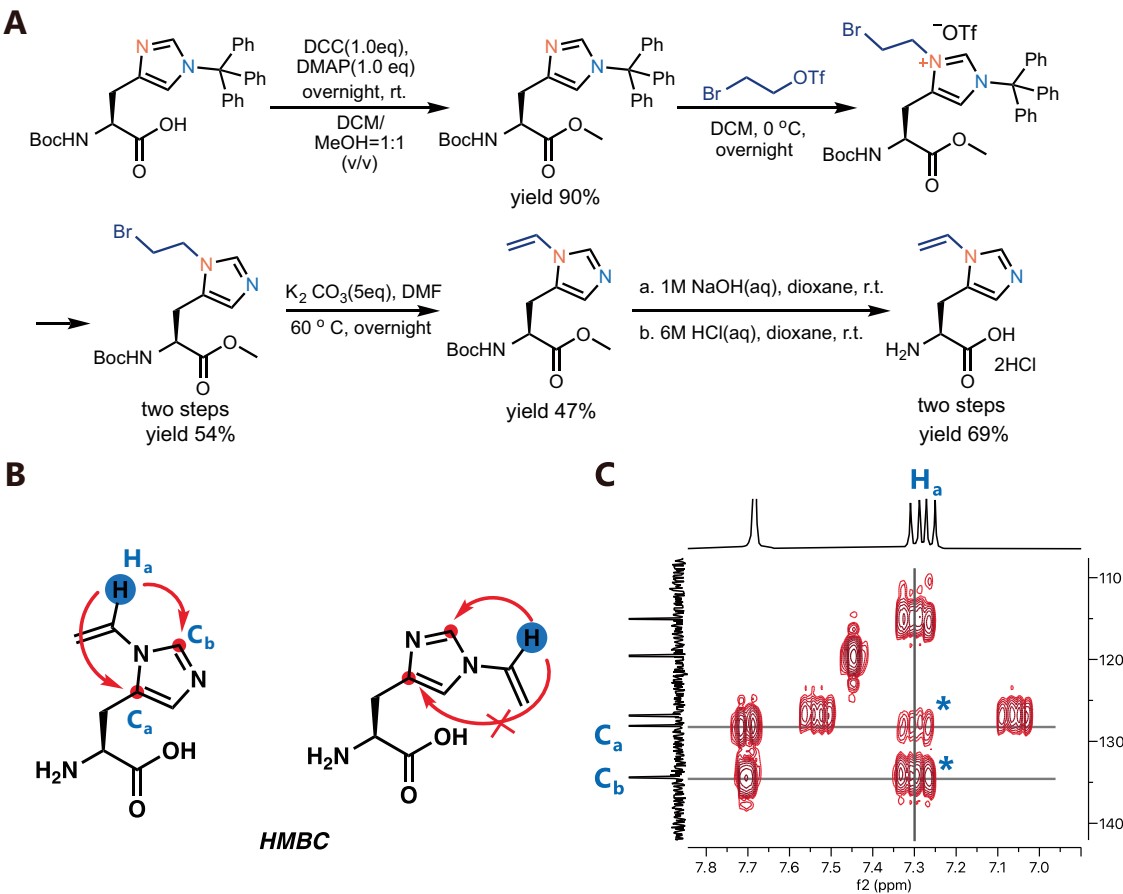

**Fig. 2 | Synthesis and characterization of δVin-H. A** The synthetic route of δVin-H. **B** The structural C-H interaction in the HMBC analysis of δVin-His and εVin-His. **C** 2D NMR(HMBC) analysis of δVin-H to confirm the substitutional position of the vinyl group on His. Source data are provided as a Source Data file.

could incorporate δVin-H into GFP-D190TAG at a similar expression level as that of GFP-WT (Fig. 3C, Supplementary Fig. 3). Even at a low concentration of δVin-H (0.1 mM), the expression level of GFP-δVin-H reached 50% of that of GFP-WT (Fig. 3D). We then named this PylRS-hit 3# for δVin-H-RS and successfully scaled up the expression of GFP-D190-δVin-H, demonstrating a remarkable yield of 40 mg/L in the DH10b strain of *E. coli* (Fig. 3D). Additionally, the molecular weight (Mw) of GFP-D190-δVin-H was verified to be consistent with the calculated Mw, indicating the stability of vinyl-imidazole under physiological conditions and during protein expression. Importantly, we demonstrated the efficient incorporation of δVin-H into the POI in mammalian cells (Fig. 3D, Supplementary Fig. 4). In summary, the results presented above highlight the development of a highly efficient and low-background δVin-H encoding system, marking a significant advancement in the field.

**Reconstruction of the catalytic center of the esterase (OE1.3) by δVin-H**

With this δVin-H-RS in hand, we sought to use δVin-H to replace the catalytic histidine in esterase, using the OE1.3 esterase as a representative example (Fig. 4A). Building upon previous work by Anthony Green and coworkers, who demonstrated that introducing δMe-H into the catalytic center significantly enhances the hydrolysis reaction activity of a computationally designed enzyme (BH32)[10,45], we aimed to expand the reaction scope and improve the catalytic performance of the OE1.3 esterase via δVin-H. In the catalytic mechanism of OE1.3 esterase (Supplementary Fig. 6A, B), the Nε of histidine acts as a nucleophilic group to attack the ester group, and the rate-determining step involves acyl release of the acyl-imidazolium intermediate. Based on this mechanism, the protonation percentage of the imidazole ring is

the key factor in determining the nucleophilicity and hydrolysis activity. Considering the pKa values of imidazole (pKa = 7.04), methyl-imidazole (pKa = 7.07), and vinyl-imidazole (pKa = 5.71), we envisioned that the δVin-H in the catalytic center will be much less protonated than δMe-H and histidine at pH=5.5, resulting in stronger nucleophilicity and higher hydrolytic activity under these physiologically relevant acidic conditions (Supplementary Fig. 6C). We then inserted δMe-H and δVin-H into esterase OE1.3 to replace the key histidine (H23). Subsequent purification of the enzyme confirmed the correct insertion of δVin-H through SDS–PAGE and mass spectrometry (Fig. 4B–C, Supplementary Fig. 5). As OE1.3 is well evolved for 2-phenylacetate substrates, we next utilized a special ester substrate (phenoxymethanol ester, Fig. 4D), which was previously reported as an endogenous esterase resistance ester[46], to further evaluated the activity of OE1.3-WT, OE1.3-δMeH, and OE1.3-δVinH under physiologically relevant acidic conditions (pH=5.5). As expected, the δVin-H catalytic center in OE1.3 exhibited the best conversion and the fastest hydrolytic reaction rate at pH 5.5 (Fig. 4D–F). Under other pH conditions, such as 6.5, 7.0, and 7.5, the ability of the three catalytic centers to hydrolyze substrates is comparable (Supplementary Fig. 7).

Furthermore, His also plays a significant role in metal binding[47], especially in the axial position of heme in various heme-containing proteins. Our subsequent goal was to use δVin-H to replace the axial histidine in enzymes, expecting to obtain improved enzymes. Four different heme-dependent proteins were chosen for this purpose, including the full-length P450-BM3 variant (HStar)-H400[48], the peroxidase variant APEX2-H163[49], the sperm whale myoglobin variant Mb*-H93[42] and a computer-designed heme-dependent protein named dnHEM1.2-H149[50]. Encouragingly, the incorporation efficiency of δVin-H was satisfactory for all four proteins, confirming the general

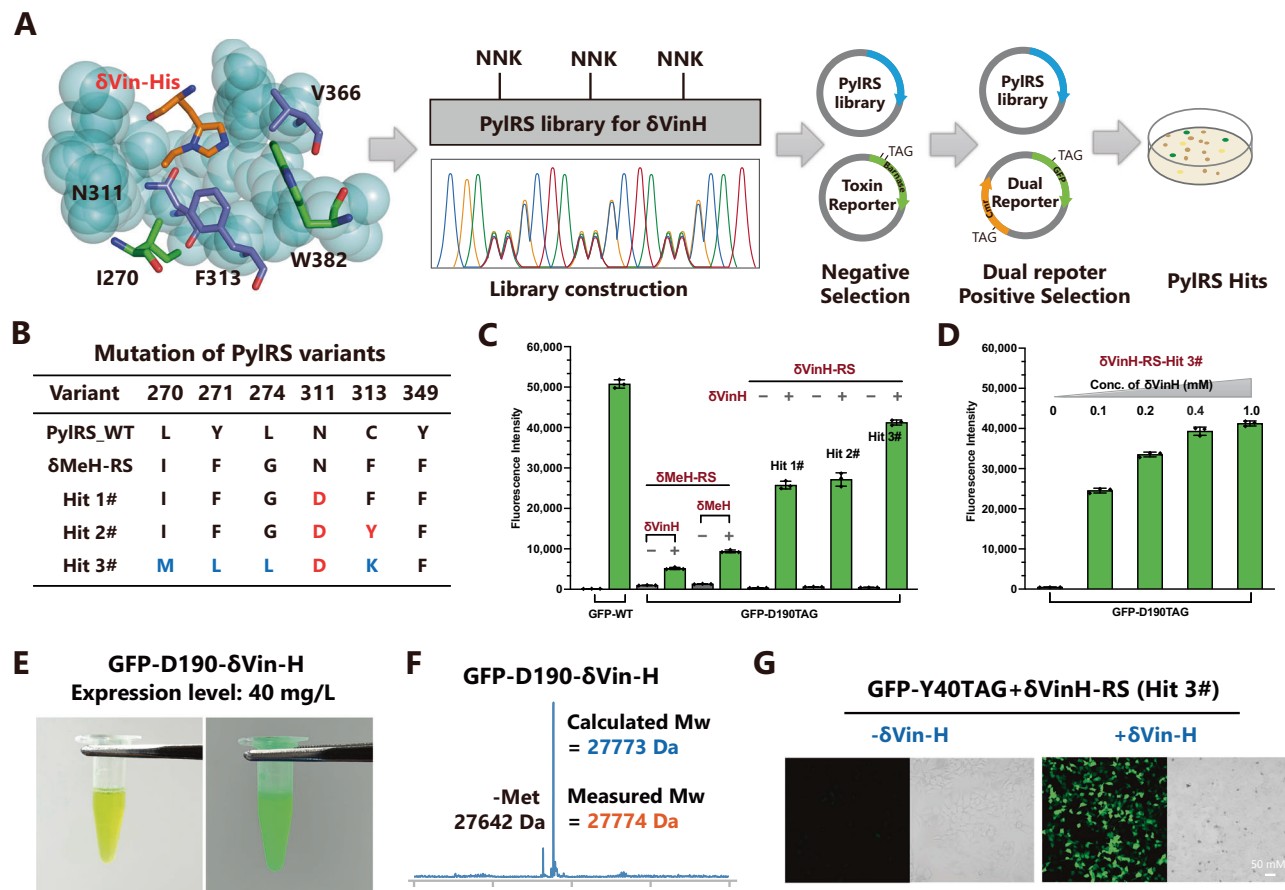

**Fig. 3 | Evolution of PylRS for the genetic encoding of δVin-H. A** The complex structure of δMeH-RS and AMP-δVin-H. The structure was regenerated from the pdb file 3qtc. **B** Evolved δVin-H-RS mutant for the recognition of δVin-H. **C, D** Validation of the selected variants through amber codon suppression of GFP-D190TAG in DH10B *E. coli*. The data in **C, D** are presented as mean values ± SD (*n* = 3, independent experiments). **E** Image of purified GFP-D190-δVin-H; the expression level reached 40 mg/L. **F** Validation of the δVin-H-incorporated GFP via LC–MS. **G** Images of the incorporation of δVin-H into GFP-Y40TAG in HEK 293 T cells. Source data are provided as a Source Data file.

applicability of δVin-H in regulating the axial coordination of heme-dependent proteins (Fig. 5A, D, Supplementary Fig. 8).

**Evolution of the carbene transferase (Mb*) by δVin-H**
We used the myoglobin variant (Mb*) protein as an example for further study[42]. Mb is a heme-dependent protein (Fig. 6A) and acts as an O₂ carrier protein in vivo[51]. In previous work, Rudi and coworkers successfully transformed this protein into a catalyst for various radical transfer reactions[42,52–55]. We first focused on the cyclopropanation of styrene with EDA as a model system to study the effect of δVin-H replacement at the axial position of heme.

Comparing the ultraviolet–visible (UV–Vis) spectra of purified Mb*-WT and Mb*-H93-δVin-H, the two proteins exhibited almost the same pattern in the resting state (Fig. 6B, Supplementary Tables 1, 2). After normalizing their absorption at 412 nm, the UV–Vis spectra of both proteins showed a redshift in the main absorption peak after the addition of the reductant (sodium dithionite). However, the reduction peak of Mb*-δVin-H was significantly greater than that of Mb*-WT. In addition, after further addition of substrate (EDA), the reduction peak of Mb*-δVin-H decreased dramatically, indicating denitrogenation and coordination of ethyl diazoacetate (EDA) to the heme structure. Moreover, the change in the reduction peak in Mb*-WT is non-significant. These results indicated that Mb*-δVin-H is more reactive than Mb*-WT in both reduction processes and carbene coordination and will show better catalytic performance than Mb*-WT.

Subsequently, we employed a cyclopropanation reaction involving styrene and EDA under reducing conditions to assess and compare the catalytic performance of Mb*-δVin-H and Mb*-WT. As shown in Fig. 6C and Supplementary Table 3, the yield of Mb*-δVin-H-catalyzed cyclopropanation of styrene reached an impressive 80% under the aerobic condition of 0.1% mol of the enzyme, while Mb*-WT yielded only 30% under the same conditions (Fig. 6C, Supplementary Figs. 9, 10). These findings illustrate that the performance of the Mb enzyme activity has been greatly improved by replacing the axial ligand of the heme from His to δVin-H under aerobic conditions. Similar improvements in enzyme activity were also observed for the electron-rich styrene substrate (Fig. 6D, Supplementary Figs. 9, 10). Given that electron-deficient styrene poses a challenge in this cyclopropanation reaction[33], we further explored the 2,3-difluorostyrene substrate for Mb*-δVin-H, revealing significantly improved conversion (Fig. 6E, Supplementary Figs. 9, 10).

Encouraged by the heightened enzyme activity of Mb in the cyclopropanation reaction with δVin-H, we anticipate that Mb-δVin-H holds potential for other carbene transfer reactions. We, therefore, utilized a silane compound as the carbene acceptor to develop the Mb*-catalyzed silane synthesis method. Upon testing, Mb*-δVin-H also exhibited better activity than Mb*-WT in catalyzing the reaction of EDA with silane, with a 33% yield without further evolution (Supplementary Fig. 11). Furthermore, we are optimistic that the performance of the Mb enzyme can be further enhanced through additional directed

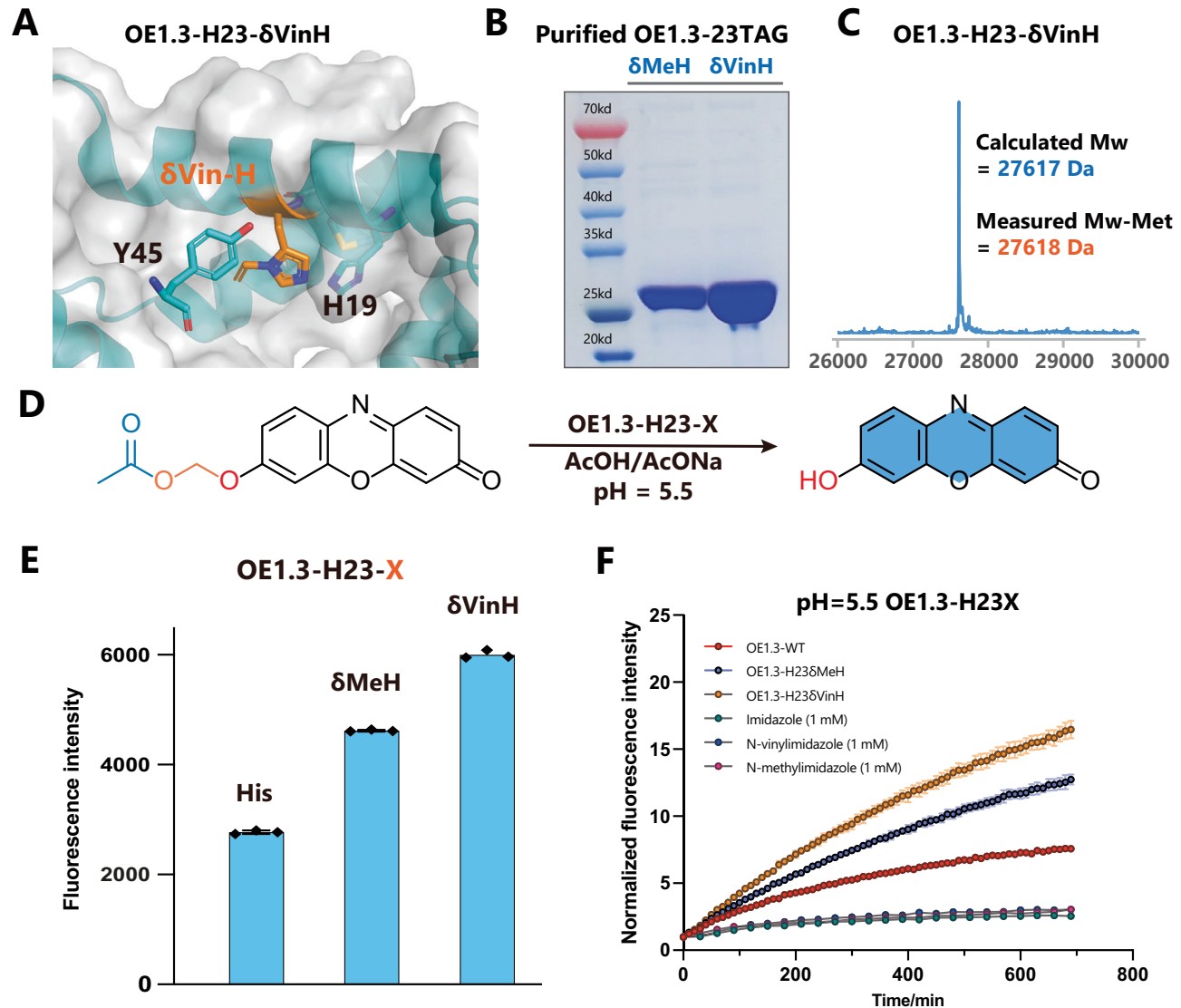

**Fig. 4 | The incorporation of δVin-H improved the activity of OE1.3 under acidic conditions. A** The structure of OE1.3-δVin-H was regenerated from the pdb file 6q7o. **B, C** Expression and purification of OE1.3-δMeH/δVin-H and LCMS validation of OE1.3-δVin-H. The data of Fig. 4B is the representative data from similar results after three independent experiments. **D–F** Comparison of the activities of OE1.3-His, δMeH, and δVin-H at pH = 5.5. The data in **E**, **F** are presented as mean values ± SEM (n = 3, independent experiments). Source data are provided as a Source Data file.

evolution. This finding suggests exciting prospects for expanding the application of δVin-H in diverse enzymatic transformations, demonstrating its versatility and potential in synthetic chemistry.

## Discussion

Histidine plays significant roles in various enzyme catalytic centers, and modifying the substitutional group of the imidazole ring can effectively restore the catalytic center, imparting new properties to biocatalysts. An illustrative example is the pivotal role of histidine as an axial ligand of heme. Through further derivatization of histidine and anchoring diverse histidine analogs into the axial ligand of heme, both the electrical properties of the metal center and the chemical environment of the active pocket can be reconstructed. This innovative approach allows for the evolution of new enzymatic conversions in heme-dependent proteins.

In essence, considering the unique effects of substitution on the imidazole ring, we believe that the continued development of histidine analogs will serve as a powerful tool in designing artificial enzymes and enhancing their activities. By expanding the chemical diversity of

catalytic histidine and exploring its applications in various enzyme systems, we will open avenues for the creation of tailored biocatalysts with improved performance and novel functionalities. This work contributes to the broader field of enzyme engineering, offering valuable insights into the fine-tuning of catalytic centers for enhanced biocatalytic applications.

## Methods

### Materials

All chemical reagents were analytical grade, obtained from Shanghai Bide Pharmatech Co., Ltd., Sigma, and used without further purification. Enzymes were obtained from New England Biolabs (NEB) and Vazyme. The primers used were obtained from Sangon Biotech (Shanghai) Co., Ltd. Antibodies were purchased from Cell Signaling Technology (anti-HisTag 12698, anti-mouse 7076 S, and anti-rabbit 7074 S) and Solarbio (anti-StrepTagII K200012M). Lipofectamine 3000 (transfection reagent) was purchased from Invitrogen. Protected Histidine **1** (Boc-His(Trt)-OH, CAS: 32926-43-5, BD33235) is commercially available from Bide Pharmatech Co., Ltd.

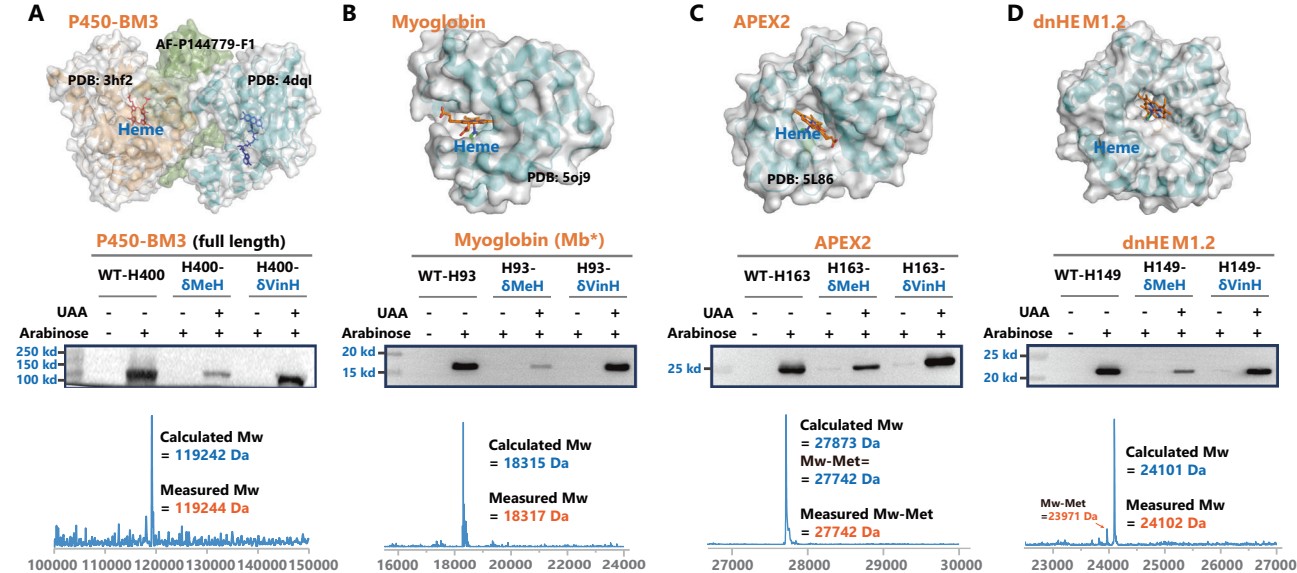

**Fig. 5 | Genetic encoding of δVin-H into different heme-dependent proteins.** Expression levels and molecular weights of four heme-dependent proteins, including **A** P450-BM3-H400, **B** APEX2-H163, **C** Myoglobin-H93, and **D** dnHEM-H149. The structure was regenerated from the pdb files: 4dql, 3hf2, and AF-P14779-F1 for P450, 5oj9 for myoglobin, 5L86 for APEX2, and 8c3w for dnHEM1.2. The data of **A**–**D** are the representative data from similar results after three independent experiments. Source data are provided as a Source Data file.

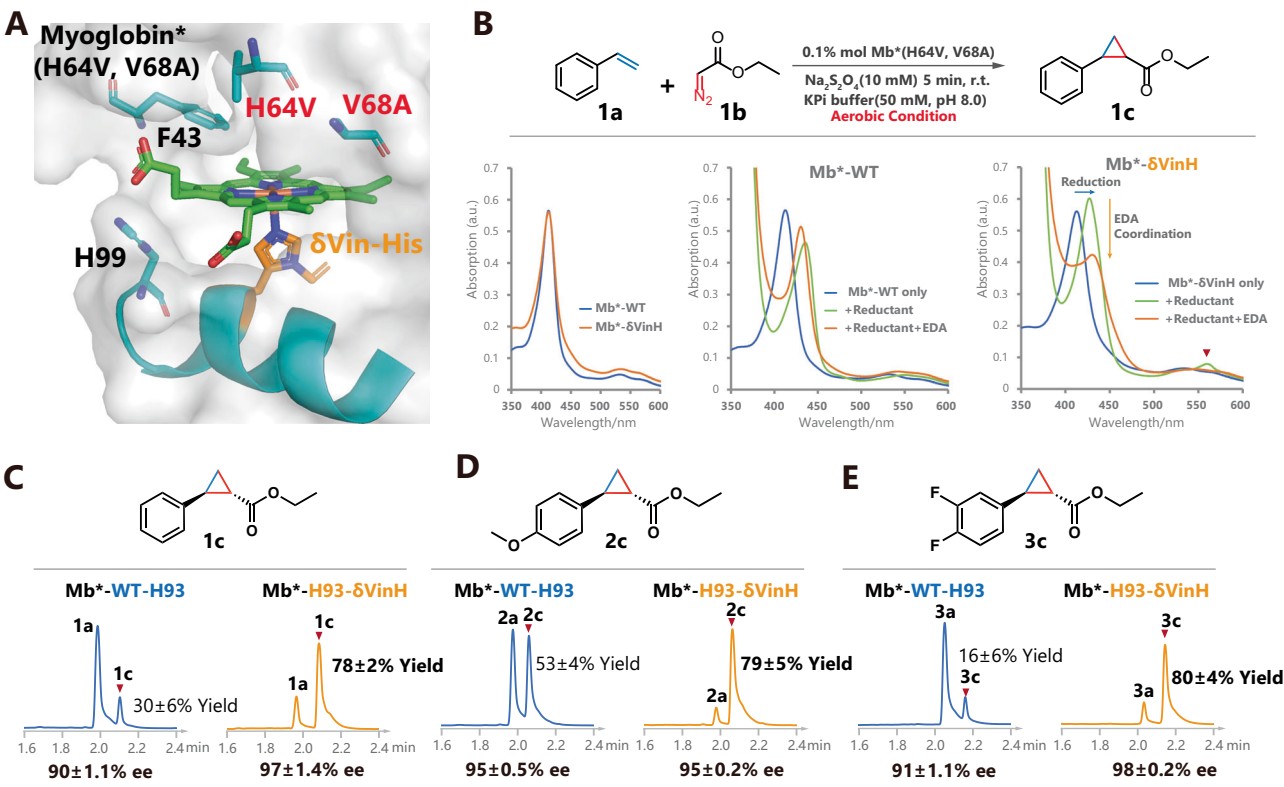

**Fig. 6 | Evolution of myoglobin via the site-specific incorporation of δVin-H.** **A** The structure of myoglobin-δVin-H regenerated from pdb file 5oj9 (illustrative purposes only and does not represent the actual structure). **B** The cycloaddition reactivity of myoglobin is improved by replacing the axial coordinate His with δVin-H. **C**–**E** The improvement in the activity of myoglobin by δVin-H was demonstrated in both electron-deficient and electron-rich styrene substrates. The yields in **C**–**E** are presented as mean values ± SD ($n = 3$, separate experiments) and the liquid chromatogram spectrum is a representative result of these three replicates. Source data are provided as a Source Data file.

## Instruments

$^1$H NMR, $^{13}$C NMR, and $^{19}$F NMR spectra were recorded with a Brucker AV 400 or 600 spectrometer at 400 or 600 MHz ($^1$H NMR), 101 or 125 MHz ($^{13}$C NMR) and 376 MHz ($^{19}$F NMR). Multiplicities were given as s (singlet); d (doublet); t (triplet); q (quartet); dd (doublets of doublet); m (multiplets), etc. High-resolution mass spectra (HRMS) were recorded on an IonSpec FT-ICR mass spectrometer with ESI. Protein purification was performed on an AKTA GO system (Cytiva). Coomassie-stained SDS–PAGE gels were imaged on a Tanon-MINI Space 3000. Western blots were imaged on a Tanon-5200SF using HRP-labeled secondary antibodies. The fluorescence intensity was monitored in a Multi-Mode Microplate Reader (Tecan Spark). UV–Vis absorption spectra were acquired on an Agilent Cary 60 instrument. All UPLC–MS analyses were performed on an ACQUITY UPLC H-Class system (Waters) equipped with an SQD2 MS detector. All UPLC analyses were performed on an ACQUITY UPLC H-Class system (Waters) equipped with an ACQUITY Premier CSH phenyl-hexyl column (1.7 µm, 2.1 × 50 mm) or Bioresolve RP mAb Polyphenyl column (2.7 µm, 2.1 × 50 mm) and a photodiode array (PAD) detector. Fluorescence imaging was performed on a Zeiss LSM 900 microscope (White Plains, NY, USA).

## Statistics and reproducibility

The uncropped and unprocessed scans of the blots are supplied in the source data file. All the DNA sequences of the protein in this manuscript are provided in the supplementary Table 6.

## GFP amber suppression assay

To assay the efficiency of amber suppression enabled by different PylRS variants, *E. coli* DH10b cells were co-transformed with pBAD/pPOS (carrying the GFP-D190TAG) and the respective pSupAR/pBK (carrying the chPylRS) plasmids. For each PylRS variant, three single clones from the transformation were used to inoculate 1 ml LB medium supplemented with 37 µg/mL chloramphenicol, and 100 µg/mL ampicillin in three separate wells of a 96-deep well plate. Plates were incubated overnight at 37 °C in 100% humidity at 500 rpm. 10 µL of the overnight cultures were used to inoculate 1 ml LB medium (1:100 dilutions) containing 37 µg/mL chloramphenicol, 100 µg/mL ampicillin, and either supplemented with the corresponding concentration of UAA (1 mM, 0.8 mM, 0.4 mM, 0.2 mM, 0.1 mM) or, as a negative control, lacking further additives. Cultures were incubated for approximately 2.5 h at 37 °C in 100% humidity and 500 rpm (OD600 = 0.5–0.6) before gene expression was induced by the addition of L-arabinose (0.2% final). Plates were further incubated overnight at 30 °C in 100% humidity at 500 rpm. On the next day, cells were washed twice with 1 mL PBS before being resuspended in the same volume. 200 µL of the suspensions were transferred to a transparent 96-well microtiter plate, and cellular GFP fluorescence intensity was measured at 510 nm upon excitation at 410 nm. 200 µL of the same suspension was used to measure the OD600 of the culture in a microtiter plate assay. GFP fluorescence was normalized to the OD600 of the respective culture.

## PylRS library construction

For library construction, full randomization was introduced by inverse PCR, and the PCR product was digested with DpnI and BsaI. Next, the PCR products were ligated to analogously digested acceptor vectors pBK vector. The ligation mixtures were purified with the Monarch® PCR & DNA Cleanup Kit (NEB) and transformed into electrocompetent DH10B. The transformants were recovered in 1 mL SOC medium at 37 °C for 1 h and plated on kanamycin-containing LB-agar plates. All the resulting colonies were washed off from the plates by using LB medium and extracted to obtain the pBK-chPylRS-Library#1/2 plasmids. The transformation yielded a library of approximately 10$^7$ mutants.

## Selection of Nδ-Vinyl Histidine specific aminoacyl-tRNA synthetases

*Round 1:* To select against synthetases that directly incorporate NAAs in response to the amber codon, *E. coli* DH10B harboring the negative selection plasmid (pNEG-Barnase) was transformed with the pBK-chPylRS-Lib#1. The transformants were recovered in 1 mL SOC medium at 37 °C for 2 h and plated onto an LB-agar plate (24 cm × 24 cm) supplemented with 0.2% arabinose, 25 µg/mL kanamycin, and 50 µg/mL ampicillin. The plates were incubated for 36 h at 37 °C. All the resulting colonies were washed off from the plates by using LB medium and extracted to obtain plasmids (pNEG-Barnase & pBK-chPylRS-Lib#1*). The synthetase plasmids were resolved from the negative selection plasmid by agarose gel electrophoresis and extracted using the Qiagen gel purification kit. Plasmids (pBK-chPylRS-Lib#1*) isolated in this negative selection were used to transform DH10B containing dual reporter positive selection plasmid pPOS-Cam-GFP. The transformants were recovered in 1 mL SOC medium at 37 °C for 1 h and plated onto an LB-agar plate (10 cm × 10 cm) supplemented with 0.2% arabinose, 1 mM δVin-H, 50 µg/mL kanamycin, 100 µg/mL ampicillin, and 20 µg/mL chloramphenicol. The plate was incubated for 12 h at 37 °C, and subsequently at 30 °C for 48 h. After incubation, the clones with green fluorescence in the plate were picked and expressed in the presence or absence of 1 mM of δVin-H. The green fluorescence measurements were carried out as described above (GFP amber suppression assay). Ten clones with the most prominent UAA-dependent GFP fluorescence were sent for sequencing. *Round 2:* *E. coli* DH10B harboring the dual reporter positive selection plasmid (pPOS-Cam-GFP) was transformed with the pBK-chPylRS-Lib#2. The transformants were recovered in 1 mL SOC medium at 37 °C for 1 h and plated onto an LB-agar plate (10 cm × 10 cm) supplemented with 0.2% arabinose, 1 mM δVin-H, 50 µg/mL kanamycin, 100 µg/mL ampicillin, and 20 µg/mL chloramphenicol. The plate was incubated for 12 h at 37 °C, and subsequently at 30 °C for 48 h. After incubation, the clones with green fluorescence in the plate were picked and expressed in the presence or absence of 1 mM of δVin-H. The green fluorescence measurements were carried out as described above (GFP amber suppression assay). Ten clones with the most prominent UAA-dependent GFP fluorescence were sent for sequencing.

## Expression and purification of proteins

We take the GFP-D190TAG-HisTag as an example and other proteins with his tag (dnHEM1.2-H149TAG-HisTag, P450-BM3-HStar-H400TAG-HisTag, Myoglobin*-H93TAG-HisTag, APEX2-H163TAG-HisTag) were purified by the same procedure.

For the expression of GFP-δMe-H and GFP-δVin-H, the plasmids pSupAR-chPylRS-PylT$_{CUA}$ and pBAD-GFP-D190TAG-6× His were co-transformed into *E. coli* DH10B cells. The cells were subsequently grown in LB media supplemented with ampicillin (50 mg/L) and chloramphenicol (34 mg/L) overnight at 37 °C. After 1:100 dilutions in 1 L of LB medium containing ampicillin (50 mg/L) and chloramphenicol (34 mg/L), the cells were left to grow at 37 °C to an OD600 of 0.5–0.6, at which point 1 mM δVin-H was added to the culture (at the final concentration). The bacteria were grown at 37 °C for another 30 min before the addition of 0.2% arabinose (final concentration) and grown at 30 °C for 24 h. The cells were then collected by centrifugation (6000 × $g$, 20 min) followed by resuspension in lysis buffer (50 mM Tris-HCl, pH 8.0, 300 mM NaCl). After sonication and centrifugation (15,000 × $g$, 30 min), the lysate was loaded onto a Ni-NTA column (Histrap 5 mL, Cytiva), which was washed with 30 mL of washing buffer (50 mM Tris-HCl, pH 8.0, 300 mM NaCl, 20 mM imidazole) and eluted with elution buffer (50 mM Tris-HCl, pH 8.0, 300 mM NaCl and 300 mM imidazole) to yield target proteins carrying δVin-H. The target proteins were finally desalted in PBS.

**OE1.3-H23TAG-StrepTagII.** For the expression of OE1.3-δMe-H and OE1.3-δVin-H, the pSupAR-chPylRS-PylT$_{CUA}$ and pBAD-OE1.3-H23TAG-StrepTagII plasmids were co-transformed into *E. coli* DH10B cells. The

cells were subsequently grown in LB media supplemented with ampicillin (50 mg/L) and chloramphenicol (34 mg/L) overnight at 37 °C. After 1:100 dilutions in 1 L of LB medium supplemented with ampicillin (50 mg/L) and chloramphenicol (34 mg/L), the cells were left to grow at 37 °C to an OD600 of 0.5–0.6, at which point 1 mM UAAs (δMe-H or δVin-H) was added to the culture (final concentration). The bacteria were grown at 37 °C for another 30 min before the addition of 0.2% arabinose (final concentration) and grown at 30 °C for 24 h. The cells were then collected by centrifugation ($6000 \times g$, 20 min) followed by resuspension in lysis buffer (100 mM Tris-HCl, pH 8.0, 150 mM NaCl, 1 mM EDTA). After sonication and centrifugation ($15,000 \times g$, 30 min), the lysate was loaded onto a Strep-Tactin column (StrepTrap XT 1 mL, Cytiva), which was eluted with elution buffer (100 mM Tris-HCl, pH 8.0, 150 mM NaCl, 1 mM EDTA, and 50 mM biotin) to yield the target proteins. The target proteins were finally desalted in PBS.

For the expression of OE1.3-WT, the pBAD-OE1.3-WT-StrepTagII plasmid was individually transformed into DH10b *E. coli*. For the expression of Myoglobin-WT, the plasmid pBAD-Myoglobin-WT-6 × His was used to individually transform DH10b *E. coli*. Next, for both of them, the transformed cells were plated onto an LB-agar plate containing 50 μg/mL ampicillin. A single colony of freshly transformed cells was cultured for 12 h in 10 mL of LB medium containing 50 μg/mL ampicillin. 10 mL of the culture was used to inoculate 1 L of 2xYT medium supplemented with 50 μg/mL ampicillin. The culture was incubated for ~2 h at 37 °C with shaking at 230 rpm. When the OD600 of the culture reached ~0.5, arabinose was added to a final concentration of 0.2% and grown at 30 °C for 24 h.

The purification of OE1.3-WT-StrepTagII and the Myoglobin-WT-6× His were conducted as the same procedure described above.

## Protein mass spectrometry
MS data for GFP-D190-δVin-H, OE1.3-H23-δVin-H, Myoglobin-H93-δVin-H, APEX2-H163-δVin-H, dnHEM1.2-H149δVin-H, and P450-BM3-HStar-H400δVin-H were acquired on an ACQUITY UPLC I-Class SQD2 (Waters) with an Bioresolve RP mAb Polyphenyl column (2.7 μm, 2.1 × 50 mm, Waters). The column was held at 80 °C and the autosampler at 10 °C. The protein solutions were desalted in water by a Biospin 6 column (Bio-Rad) and subjected to LC-MS analysis within 1 hour. The final protein concentrations were adjusted to 0.5 mg/mL. Mobile phase A was 0.1% formic acid in water, and mobile phase B was acetonitrile with 0.1% formic acid. A constant flow rate of 0.2 mL/min was used. The gradient used was 5% B for 1 min, increasing linearly to 100% B for 2.5 min, holding at 100% B for 1 min, changing to 5% B in 0.2 min, and holding at 5% for 1 min. The MS data were collected on a Waters SQD2 detector with the m/z range of 500–1800. The desolvation temperature was 350 °C with a flow rate of 1000 L/h. The voltages used were 1.50 kV for the capillary and 50 V for the cone. MassLynx was used to operate the LCMS and analyze the data. The deconvolution of the protein mass spectrum data is processed by using MaxEnt with the corresponding MW range with a resolution of 1 Da/channel. The molecular weight of the protein was predicted using the ExPASy ProtParam tool (https://web.ExPASy.org/protparam/). For each purified protein, the mass analysis was conducted once.

## Coomassie brilliant blue (CBB) staining
Proteins were separated by SDS–PAGE. For CBB staining, the gel was removed from the electrophoresis chamber, and enough 0.5% Coomassie Blue R-250 (prepared in 50% methanol/10% acetic acid) was added to cover the gel for 1–2 h until the gel was a uniform blue color. Then, the samples were destained with 40% methanol and 10% acetic acid, and the solution was replaced every 10-20 min until faint bands were observed.

## Western blotting
Proteins were separated by SDS–PAGE. For western blotting, proteins were transferred to PVDF membranes. The membranes were blocked for 60 min with gentle agitation using 5% dry milk in TBST. The membranes were subsequently incubated with the indicated primary antibodies overnight at 4 °C. Excess primary antibody was removed, and the membranes were washed with TBST (3 × 5 min). To visualize the primary antibodies, the membranes were incubated with the appropriate secondary antibodies for 1 h at room temperature. The membranes were washed with TBST (3 × 5 min) and imaged.

## Mammalian cell culture
HEK293T cells were obtained from the American Type Culture Collection (ATCC). HEK293T cells were grown in DMEM (Dulbecco's modified Eagle's medium) supplemented with 10% fetal bovine serum (FBS).

## Cell transfection and UAAs incorporation
HEK293T cells were cultured at 37 °C in a 5% $CO_2$ atmosphere in DMEM (Gibco) supplemented with 10% FBS and 1× pen-strep solution. The cells were plated in plates and grown to 80% confluency for transfection. Cells were transiently transfected with Lipofectamine 3000 (Invitrogen) according to the manufacturer's protocol. Double transfections were performed using equal amounts of both plasmids (PylRS-PylT$_{CUA}$ plasmid and POI-Amber plasmid). Before transfection, the medium was replaced with fresh medium supplemented with the UAAs. Subsequently, the cell culture medium was exchanged for this mixture, and the cells were cultured for another 20 h before the subsequent experiments.

## Fluorescence imaging
HEK293T cells were grown to 80% confluency for transfection in six-well plates. Plasmids encoding the GFP-Y40TAG-FLAG mutants were cotransfected with the plasmid bearing the PylRS-PylT$_{CUA}$ pair into cells via Lipofectamine 3000 and cultured in the presence of 1 mM δVin-H or δMe-H for another 24 h. Next, the cells in each well were observed via fluorescence microscopy.

## UV−Vis measurements
UV–Vis absorption spectra of myoglobins were recorded in 50 mM potassium phosphate buffer (pH 8.0) at a final concentration of approximately 5 μM.

## Hemeochrome assay
To determine the heme concentration of the purified proteins and subsequently calculate their heme extinction coefficients, a pyridine hemeochrome assay was carried out as described in the literature[56]. The details of this assay are listed as follows: the purified protein was mixed with a solution containing 0.2 M NaOH, 40% (v/v) pyridine, and 0.5 mM K$_3$[Fe(CN)$_6$] at a final protein concentration of approximately 10 μM. The absorbance spectra of the oxidized proteins were recorded between 500 and 600 nm on a spectrophotometer. Subsequently, 10 μL of a solution of 0.4 M sodium dithionite was added, and spectra were recorded every minute until no further absorbance change was observed. The last band was assigned to the reduced protein. Heme concentrations were calculated from the difference between the reduced (556 nm) and oxidized (540 nm) proteins using the extinction coefficient ε(red-ox) =23.98 mM$^{-1}$·cm$^{-1}$ of the Pyr$_2$-heme b complex. The extinction coefficients of the myoglobins were calculated from the measured heme concentrations relative to the Soret band in the UV–Vis spectrum of the protein. Reported values are the average of three independent measurements. Errors are given by the standard deviation.

## General procedures for Ester hydrolysis reaction
Acetic acid/sodium acetate buffer (180 μL, pH = 5.5), enzymes (10 μL, 1 μM final concentration), and Resorufin ether (10 μL, 100 μM final concentration) were added to 96-well plates. Later, the 96-well

plate was put in a microplate reader and shaken up for 5 seconds per 3 min. The formation of the Resorufin product was monitored over 11 h by using a microplate reader in which the wavelength of excitation light is 540 nm, and the wavelength of emission light is 600 nm in the microplate reader. The concentrations of OE1.3 variants were determined by BCA. The data were collected with Tecan's Spark Control software.

## General procedures for cyclopropanation reaction

The cyclopropanation reaction was run on a 200 μL scale in 2 ml centrifuge tubes. The Mb* variants were added to 50 mM potassium phosphate buffer pH 8.0 containing sodium dithionite (final concentration: 10 mM). The reactions were initiated by adding styrene analogs (5 μl of 400 mM stock in ethanol) and EDA (5 μl of 800 mM stock in ethanol). The reactions were stirred at 500 r.p.m. using a magnetic stirring bar and quenched with 50 μl of 3 M hydrochloric acid at appropriate time points. For product quantification, 1 mL methanol was added followed by UPLC analysis (The concentrations of Mb* variants were determined by UV–Vis absorbance measurements at 412 nm).

## General procedures for Si-H insertion reaction

The Si-H insertion reactions were run on a 200 μL scale in 2 ml centrifuge tubes. The Mb* variants were added to 50 mM potassium phosphate buffer pH 8.0 containing sodium dithionite (final concentration: 10 mM). The reactions were initiated by adding silane (5 μl of 400 mM stock in ethanol) and EDA (5 μl of 800 mM stock in ethanol). The reactions were stirred at 500 r.p.m. using a magnetic stirring bar and quenched with 1 mL methanol at appropriate time points. The product was quantified by UPLC analysis. (The concentrations of Mb* variants were determined by UV–Vis absorbance measurements at 412 nm).

## Density functional theory (DFT) calculations

DFT calculations were performed at the B3LYP-D3BJ/6-311 G(d, p)[57] level using the Gaussian09 computational chemistry package. The IEF-PCM solvent model[58] for solvation by water was used for the computations. The Cartesian coordinates (Å) for the DFT-optimized structures are listed in the Supplementary Table 5.

## Reporting summary

Further information on research design is available in the Nature Portfolio Reporting Summary linked to this article.

# Data availability

The data generated in this study are provided within the paper and in the Supplementary Information. Source data are provided in this paper.

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

## Acknowledgements

This work was supported by the National Key Research and Development Program of China (2021YFA0910800, 2023YFA1506500), the National Natural Science Foundation of China (22077059, 22277048, 92253304), Beijing National Laboratory for Molecular Sciences (BNLMS202012), the Young Elite Scientists Sponsorship Program from the China Association for Science and Technology (YESS20200010), Shenzhen Medical Research Fund (A2303067) and Shenzhen Science and Technology Program (JCYJ20210324104210028, RCYX20210609103118010, KQTD20221101093558015).

## Author contributions

H.H., T.Y., and C.L. contributed equally. J.W. and H.H. conceived the study. H.H. conducted most of the experiments unless otherwise specified. T.Y. contributed to the Myoglobin-catalyzed cyclopropane reaction and Si-H insertion reaction. C.L. contributed to the ester hydrolysis reaction and biochemical experiments. Y.L. contributed to the chemical synthesis of A-Me-Res and the ester hydrolysis reaction. Z.W. contributed to the fluorescent imaging. X.W. contributed to biochemical experiments. J.W. and H.H. wrote the paper with inputs from all authors.

## Competing interests

The authors declare no competing interests.
