## [Peer Review File · Nature Communications]

Genetically Encoded N δ -Vinyl Histidine for the Evolution of Enzyme Catalytic CenterEditorial Note: This manuscript has been previously reviewed at another journal that is not operating a transparent peer review scheme. This document only contains reviewer comments and rebuttal letters for versions considered at *Nature Communications*. Mentions of the other journal have been redacted.

REVIEWERS' COMMENTS

Reviewer #2 (Remarks to the Author):

I reviewed this manuscript when it was submitted to [redacted]. I commended the authors for improving the figures and legends substantially so they are clearer to readers, and that information on replicates and variance in data is now clearly provided. The application of N-vinyl histidine to myoglobin is now performed with the best Mb variant for cyclopropanation, with N-vinyl histidine incorporation capable of improving the activity of Mb* further. This addressed my key concern from the previous round, and demonstrated that N-vinyl histidine incorporation can provide an added benefit to enzyme variants obtained from traditional mutagenesis.

From the previous round of review, I find the directed evolution part of the manuscript to already be very satisfactory. The evolution experiments are well-designed and well-performed, and the obtained mutants, particularly Hit 3, demonstrated great incorporation activity and selectivity for N-vinyl histidine.

I recommend publication of the manuscript, which is a complete proof-of-concept for the development of a histidine analog with a modified pKa and its incorporation into multiple useful proteins via genetic code expansion, in *Nature Communications*.

[**Editorial note:** Reviewer 2 also assessed author's responses to concerns of Reviewer 1 and considers that all questions were answered satisfactorily.]